# Development of a Decisional Procedure Based on Fuzzy Logic for the Energy Retrofitting of Buildings

**Linda Barelli** , **Elisa Belloni \*** , **Gianni Bidini, Cinzia Buratti** and **Emilia Maria Pinchi**

Department of Engineering, University of Perugia, 06125 Perugia, Italy; linda.barelli@unipg.it (L.B.);
gianni.bidini@unipg.it (G.B.); cinzia.buratti@unipg.it (C.B.); emilia7@hotmail.it (E.M.P.)
**\*** Correspondence: belloni.unipg@ciriaf.it; Tel.: +39-075-5853562

**Abstract:** This paper concerns the development of an automatic tool, based on Fuzzy Logic, which is able to identify the proper solutions for the energy retrofitting of existing buildings. Regarding winter heating, opaque and glazing surfaces are considered in order to reduce building heat dispersions. Starting from energy diagnosis, it is possible to formulate retrofitting proposals and to evaluate the effectiveness of the intervention considering several aspects (energy savings, costs, intervention typology). The innovation of this work is represented by the application of a fuzzy logic expert system to obtain an indication about the proper interventions for building energy retrofitting, providing as inputs only few parameters, with a strong reduction in time and effort with respect to the software tools and methodologies currently applied by experts. The novelty of the paper is the easy handling properties of the developed tool, which requires only a few data about the buildings: not many such methods were developed in the last years. The energy requirements for winter heating before and after particular interventions were evaluated for a consistent set of buildings in order to produce the required knowledge base for the tool's development. The identified appropriate inputs and outputs, their domains of discretization, the membership functions associated to each fuzzy set, and the linguistic rules were deduced on the basis of the knowledge determined in this was. Therefore, the system was successfully validated with reference to further buildings characterized by different design and architecture features, showing a good agreement with the intervention opportunities evaluated.

**Keywords:** building design and architecture; energy retrofitting; energy efficiency in buildings; energy diagnosis; fuzzy logic

## 1. Introduction

### 1.1. General Overview of Building Energy Retrofitting

Considering the continuous increase in energy consumption typical of the building sector, the present work proposes a tool for the automatic identification of the proper solutions for energy retrofitting [1–6] in order to increase the efficiency of existing buildings in reference to the winter heating. The building sector significantly contributes to environmental pollution through the extensive exploitation of territory, the use of non-renewable resources, and the large amount of energy required throughout a building's life cycle. The construction sector in Europe is indeed responsible for about 40% of the energy consumption and, consequently, for high $CO_2$ emissions [7–12]: it represents a sector with a high potential of reduction, and several measures have been taken by the European Commission to reduce this critical issue, as considered also in the recent European Directive 2012/27/EU [13–15]. Energy consumption has to be reduced in both new and existing buildings, but especially the latter need an effective energy renovation, according to the new directives: in particular, all countries have had to retrofit 3% of their public buildings per year since 2014. Moreover, from an economic point of view, residential building operating costs are significant when compared to those for construction; consequently, energy

saving measures should be implemented in new building construction, as is also indicated by the current European and national regulations, in order to achieve maximum efficiency. Anyway, it is highlighted as new buildings are only a little portion of the total. In the specific Italian scenario, characterized by 40% of domestic energy consumption being due to the housing sector [16,17], about 2/3 of the existing buildings were built prior to 1976, when the first national law, i.e., No. 373/76 [18], on the energy performance of buildings was issued. Consequently, the energy refurbishment of existing buildings represents a crucial issue. It is clear that, aiming for consumption reduction, it is first necessary to improve the building envelope by increasing the insulating performance. Most of the energy consumption is due to heating and cooling, whereas the remaining is related to hot water production, household appliances and lighting. Moreover, contrary to the case of heating plants, nowadays in Italy not all the buildings are equipped with cooling plants, even if the use of cooling devices in the last few years was largely widespread. For this reason, in the present paper, the attention was focused on the energy savings from building refurbishment related to the winter heating. In the future, a similar study will be carried out focusing on the summer case.

The aim of the preliminary analysis was to identify the factors on which to intervene, aiming to maintain or improve the level of indoor comfort with lower consumptions. Starting from an energy audit, it is then possible to formulate action proposals which are beneficial for both economic and energy (potential savings) aspects [19–21]. There are many retrofitting measures that can be applied to each building, and selecting the most appropriate ones is not an easy task. The absence of adequate thermal insulation is considered to be the main cause of heat loss. The contribution of the external walls [22–25], the effect of opaque vertical elements, the roof (the dispersion through an un-insulated roof can represent more than 25% of the total losses of a building) and the glazing surfaces is especially highlighted in this study. In particular, windows represent the main structural elements responsible for the largest heat loss in winter (depending on the U-value parameter) and significant heat entrance in summer (depending on the solar factor, g) [26,27].

*1.2. State of the Art in Artificial Neural Network and Fuzzy Logic Tools*

The studies already available in the literature mainly focus on the calculation and prediction of buildings' energy consumptions, also by applying Artificial intelligence (AI) techniques. Regarding the energy requirement calculations, engineering methods, statistical methods, or AI methods [28–32] are applied. In Castro et al. [33], a decision matrix was proposed as a tool to identify the most appropriate retrofit measures of an existing building. The matrix was calculated by using dynamic simulation tools; the energy deviations produced by modifying the input variables were quantified and a final sensitivity analysis was carried out. The outputs obtained by this decision matrix were the building loads of each retrofit measure and the associated cost. In order to facilitate the selection of the optimal retrofit actions, Rosso et al. [34] proposed the application of an active archive non-dominated sorting genetic algorithm (aNSGA-II) geared towards multi-objective optimization. The results of the algorithm's implementation were analyzed with respect to a residential building located in Rome, Italy. Regarding energy consumption prediction, AI methods, such as artificial neural networks (ANNs) [35–40], are widely used instead to provide hourly load profiles. Moreover, AI techniques are also applied in the building sector to characterize materials, i.e., for the estimation of their thermal diffusivity. In this regard, Roman et al. [40] presented a comprehensive and in-depth systematic review of the up-to-date literature related to the application and characterization of ANN-based metamodels for building performance simulations. Moreover, in [37], new and efficient approaches based on artificial neural networks and neuro-fuzzy systems are proposed. Ascione et al. [41] showed that artificial neural network predictions can allow a wide diffusion of rigorous approaches for retrofit design, which are currently hampered by the excessive computational burden. In particular, office buildings built in South Italy during 1920–1970 were investigated, and the developed artificial neural networks were shown to

be able to replace standard building performance simulation tools for the evaluation of the retrofitting techniques' effectiveness, thereby producing a substantial reduction of the computational efforts and times.

Many previous studies tried to develop multi-objective tools which are able to improve simultaneously the energy (cooling and electricity demand), comfort (thermoigrometric, visual and acoustic), environmental (considering all phases of an LCA) and economic (costs and return time) aspects [42–44]. As for example, D'Amico et al. [42] proposed a simple and reliable tool which simultaneously solves the energy and environmental balance of buildings. An energy database created in previous works, representative of the Italian building stock, was used by the authors in order to retrieve the energy and environmental performance of non-residential buildings. Several typologies of Artificial Neural Networks were analyzed, and the best results were selected by a deep statistical analysis and results comparison. In [45], a new method integrating a genetic algorithm (GA), an artificial neural network (ANN), a multivariate regression analysis (MRA) and a fuzzy logic controller (FLC) was proposed in order to optimize the indoor environment and energy consumption, based on simulation results. In particular, the GA process was used to search for the optimal solution, whereas the ANN and CFD tools were used to obtain the values of the objectives for each individual. The fuzzy logic system was used to control the execution routine of the CFD process and to reduce the computational cost by up to 40. In another study [43], a building performance optimization technique was well applied in order to design energy-efficient buildings, i.e., a residential building located in the Marrakech region, Morocco. The aim was to minimize its energy demand, especially for heating and cooling, as well as to maximize the indoor thermal comfort of the two most important targets for building designers. A model with good predictive accuracy was developed by the ANN in order to target the overall research space of the two expected objectives (energy performance and indoor thermal comfort). This was accomplished by using a database of 35 samples, which were simulated by a platform developed in the TRNSYS software environment. In fact, ANNs can be applied also for detailed thermal comfort analyses, as shown in [46,47], in which was developed a surrogate model to speed up the thermal comfort prediction for any member of a building category, relating to different energy retrofitting actions and ensuring high reliability by testing the entire simulation process, with real data measured in-situ.

Only a few studies deal instead with the development of decisional tools for the building sector, especially for building refurbishment. In [48], a decisional tool based on AI techniques was proposed, but for building energy management. The developed tool helps to guarantee desirable levels of living quality, as well as energy savings, for environmental protection through intelligent monitoring and the optimized start/stop of HVAC and lighting controls. Existing buildings are, in general, constrained by old equipment, aging infrastructure and inadequate operations resources. Two artificial neural network (ANNs) models were developed by Poço et al. [49] for a retail store located in Lisbon in order to predict the building power consumption and indoor average temperature; the tool was able to verify the impact of small variations in the air-handling unit electricity consumption. Only a methodology, described in [50,51], was found for buildings refurbishment. The proposed method, developed within a project funded by the European Commission, aims to support the refurbishment or retrofitting (upgrading) of apartments. Retrofitting measures are suggested to promote energy efficiency, living conditions and structural features, mainly depending on the budget assigned by the user for the global refurbishment. Moreover, the tool addresses not only technical experts, but also owners and other operators. Exergoeconomic analysis and optimization is another common practice in sectors such as the power generation, helping engineers to obtain more energy-efficient and cost-effective energy systems design. This calculation method can be applied to typical open-source building energy simulation tools, such as EnergyPlus [52]. In [52], the enhanced simulation framework was tested considering a primary school as a case study. The results demonstrated that the proposed simulation framework provided users with thermodynamic, efficient

and cost-effective designs, even under tight thermodynamic and economic constraints, suggesting its use in everyday building energy retrofitting practice.

Other interesting fuzzy logic applications deal with lighting aspects: in particular, Chiesa et al. [53] introduced a working prototype of a fuzzy logic IoT system, which controls natural and artificial light balance in combination with a dynamic shading system. A control app was developed to allow user interaction by setting seasonal automatic modes or manual functionalities, and the control system adopted a fuzzy logic solution, which is able to ensure rapid control without high computational effort. Mattoni et al. [54] and Ilbeigi et al. [55] optimized the indoor lighting of an office by a Genetic Algorithm technique. The results indicated that in the optimum conditions, the uniformity of illuminance increased considering a reduction in the number of luminaires and the maximum Unified Glare Rating (UGR) values. Finally, a complete set of fuzzy systems to control the operation of the various parts of a building's automation was proposed by Mpelogianni and Groumpos [56]. A further work [57] implemented fuzzy logic and machine learning techniques in order to ensure the proactive energy management of a building on the basis of the first results [56].

### 1.3. Aim of the Study

Based on the previous considerations, the authors propose a methodology to support decision makers during the planning and design phases of a building. The method can provide a valid estimation of the energy performance of buildings. The lack in the literature of an automatic tool which provides intervention suggestions on building retrofitting, with reduced spent time and effort, including all building typologies, is clear. The present study proposes, therefore, a tool which is able to reduce energy consumption for indoor heating through envelope upgrading, which provides, for each specific case, the level of opportunity of particular interventions (the most effective ones). The target is to assign automatically, through quantitative criteria, higher opportunity levels to different interventions by providing higher potential energy savings (taking into account economic and feasibility qualitative criteria). The use of AI to solve this complex problem can represent a valid and attractive alternative. The implementation requires the presence of a suitable database set, such that the output data strongly relate to one or more input data. In order to develop such a decisional tool, a fuzzy logic technique was applied. Fuzzy logic also considers "nuanced" aspects, and it therefore allows us to reduce the uncertainty and inefficiency resulting from the variability of the factors typical of the phenomenon under analysis [28]. Therefore, a knowledge base consisting of particular case studies was initially created. Section 3.1 reports all of the details regarding the data gathering, and the analysis and elaboration carried out by the authors. Then, starting from this knowledge base, the tool was developed (see Section 3.2) to provide, for the generic building, the automatic evaluation of intervention opportunities (in reference to a particular set of solutions).

Finally, as detailed in Section 4, the tool's performance was validated through application to further particular case studies. This validation phase demonstrated that the proposed methodology is suitable to resolve, through an automatic tool, a complex and globally not-linear problem usually addressed by the experts of the fields. In particular, the application of a Fuzzy Logic expert system to obtain an indication about proper interventions for building energy retrofitting represents an innovation with respect the state of the art. Moreover, only a few parameters are required by the developed tool, with a strong reduction in the needed time and effort with respect to the software tools and methodologies currently applied by experts. This is the potential impact of this study, once the developed tool is upgraded, considering further intervention typologies and extending it also to the summer case. As shown above, not many Fuzzy Logic tools were developed in the last few years in the building sector, and they were generally not very useful instruments; this research work could lay the foundations to fill the lack in the current literature.

## 2. Description of the Method: Fuzzy Logic

Fuzzy logic is able to extract decisions from a base of knowledge previously acquired from the experience of another classic system which is able to perform the same task. What is therefore useful to include in this decisional system is the human ability to consider particular factors which are difficult to quantify and formalize, which by their nature are nuanced aspects of reality [58].

The knowledge base generally contains all of the information about the system, allowing us to process the input data to obtain the output. In the fuzzy logic design, from this information, it is possible to deduce three basic components: the fuzzy sets for the discretization of the variable's domain, the relative membership functions (curves that define the membership degree—between 0 and 1—of a generic variable to each set), and the linguistic rules of inference through which the output evaluation is performed. A fuzzy rule is usually expressed with an if–then construct, and may submit one or more antecedents and one or more consequents [59]. Because inputs are crisp values, it is necessary to make a conversion to translate a number in the fuzzy data; this operation is named fuzzification (Figure 1). After fuzzification, the inference process converts the input fuzzy sets into output fuzzy sets. The largest applied inference method is the one proposed by Mandami [60,61], based on the application of the minimum method (Equation (1)).

$$B \rightarrow \mu A \rightarrow B \ (uA, uB) = min \ (\mu A(uA), \mu B(uB)) \tag{1}$$

where variables uA and uB are defined respectively in the domains A and B, whereas functions μA(uA) and μB(uB) are the corresponding membership functions. Mamdani fuzzy inference was first introduced as a method to synthesize a set of linguistic control rules obtained from experienced human operators. In a Mamdani system, the output of each rule is a fuzzy set. Because Mamdani systems have more intuitive and easier-to-understand rule bases, they are well-suited to expert system applications where the rules are created from human expert knowledge. Moreover, the fuzzy inference engine also provides the aggregation of all of the outputs obtained by each rule, in order to give a single fuzzy set. Fuzzy inference is the process of formulating the mapping from a given input to an output using fuzzy logic. The mapping then provides a basis from which decisions can be made or patterns discerned. The process of fuzzy inference involves all of the pieces that are described in Membership Functions, Logical Operations, If–Then Rules and the aggregation method. The fuzzy inference process consists of several steps: the fuzzification of the input variables, the application of the fuzzy operator (AND or OR) in the antecedent phase, the implication from the antecedent to the consequent (inference phase), the aggregation of the consequents across the rules, and finally the defuzzification step.

The first step (fuzzification) is the determination of the degrees to which the inputs, always crisp numerical values, belong to each of the appropriate fuzzy sets via membership functions. Fuzzification associates to each input one or more degrees of membership (values in the 0–1 interval) in reference to the defined fuzzy sets. After the inputs are fuzzified, the degrees to which each part of the antecedent is satisfied are known for each rule. If the antecedent of a rule has more than one part, the fuzzy operator is applied to obtain one number that represents the result of the antecedent rule. Therefore, this number can be applied to the output function.

Before applying the implication method, it is necessary to determine the rule weight. Every rule has a weight (a number from 0 through 1) which is applied to the number given by the antecedent. Generally, if this weight is equal to 1, it has no effect on the implication process; it is possible to decrease the effect of one rule with respect to the others by changing its weight value to something different from 1. After the proper weighting has been assigned to each rule, the inference is implemented, usually by truncating the output set with the membership degree resulting from the antecedent. Aggregation is the process by which the truncated output fuzzy sets, returned for each rule by the inference process, are combined into a single fuzzy set. Three types of built-in aggregation methods

are the max (maximum), sum (sum of the rule output sets) and probOR (probabilistic OR) functions. The output of the aggregation process is one fuzzy set for each output variable.

The final step is the defuzzification of the fuzzy set [62] resulting from the aggregation phase cited above. The input for the defuzzification process is the aggregate output fuzzy set; the outcome for each output variable is a crisp value. There are five built-in defuzzification methods which are supported: centroid, bisector, the middle of the maximum (the average of the maximum value of the output set), the largest of the maximum, and the smallest of the maximum. Among the available strategies, the maximum and the centroid methods are highlighted. The most popular defuzzification method of the aggregated fuzzy set is the centroid calculation, which returns the abscissa of its center gravity. According to the maximum method, instead, the output value is calculated as the point where the output membership function reaches its maximum. A fuzzy inference diagram displays all of the parts of the fuzzy inference process, from fuzzification through to defuzzification (Figure 1). A general schematization of fuzzy logic is shown in Figure 2.

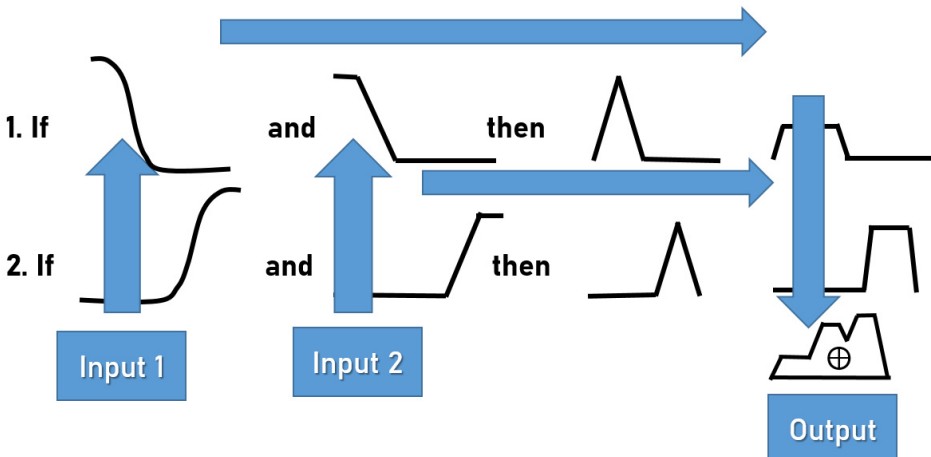

**Figure 1.** A fuzzy inference diagram.

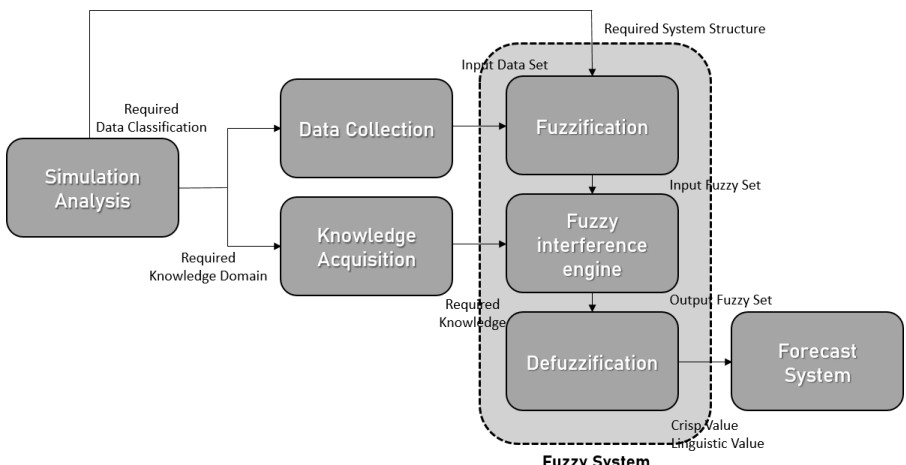

**Figure 2.** Schematization of fuzzy logic.

## 3. Design of the Fuzzy Decisional Tool

As already indicated, this study aims to develop a decisional tool based on fuzzy logic for the identification of proper interventions for building energy retrofitting; in the first instance, we analyzed the winter case, considering the main building components responsible for the greater heat losses. In particular, focusing attention on external vertical walls, roofs and glass surfaces, the decisional tool should be able to identify and discern the

typology of useful intervention which is able to improve the building envelope and, above all, to indicate a priority in case of multiple interventions. The design of such a system was carried out, as detailed in the following, by identifying appropriate inputs and outputs, their discretization, and the associated membership functions, beyond that the rules set.

### 3.1. Problem Analysis

Aiming to develop the fuzzy tool cited above, the main tasks are the inputs' definition, the evaluation of the outputs and related characteristics, and the design of the interference engine.

Therefore, the appropriate inputs among the parameters characterizing the building envelope efficiency were identified: the transmittance of both vertical and horizontal opaque walls ($U_{wall}$, $U_{roof}$), the transmittance of glass surfaces ($U_{glass}$), the percentage ratio of the glass and opaque areas (Window To Wall Ratio—WWR), and the winter energy demand (WinterEnDem). For the $U_{wall}$, $U_{roof}$ and $U_{glass}$ parameters, in the case of multiple component typologies, the weighted mean values (with weights for the corresponding areas) of the building under study were considered. Several databases were taken into account in order to evaluate the inputs' variability ranges, which were representative of the variable domains and their subdivision in several of the classes determined. For all of the input parameters, Table 1 shows the defined classes, with data about the related quantitative limits.

**Table 1.** Classes of the input variables (VVL = very very low, VL = very low, L = low, M = medium, H = high, VH = very high, VVH = very very high).

| Uwall (W/m$^2$K) | | Uroof (W/m$^2$K) | | Uglass (W/m$^2$K) | | WWR (%) | | WinterEnDem (kWh/m$^2$) | |
|---|---|---|---|---|---|---|---|---|---|
| U < 0.3 | VL | U < 0.5 | L | U < 1.4 | VL | % < 6 | VL | <10 | VVL |
| 0.3 ≥ U < 0.55 | L | 0.5 ≥ U < 0.7 | M | 1.4 ≥ U < 1.7 | L | 6 ≥ % < 10 | L | ≥10 & <30 | VL |
| 0.55 ≥ U < 0.75 | M | 0.7 ≥ U < 1 | H | 1.7 ≥ U < 2.3 | M | 10 ≥ % < 17 | M | ≥30 & <50 | L |
| 0.75 ≥ U <1.4 | H | U ≥ 1 | VH | 2.3 ≥ U < 3.4 | H | 17 ≥ % < 20 | H | ≥50 & <70 | M |
| U ≥ 1.4 | VH | | | U ≥ 3.4 | VH | % ≥ 20 | VH | ≥70 & <90 | H |
| | | | | | | | | ≥90 & <120 | VH |
| | | | | | | | | ≥120 | VVH |

In order to make clear the procedure followed, five classes were defined for the $U_{wall}$ parameter. Such a subdivision was obtained on the basis of the wall stratigraphy (depending on the construction typology). In detail, a large database of different opaque walls was initially formed, considering for each wall its stratigraphy, so as to be able to characterize the presence of particular elements with higher performance (the insertion of one or more insulating layers). Therefore, the corresponding $U_{wall}$ value was determined for each vertical opaque element. Considering the large variety of limits imposed by the different European standards, no specific limit imposed by the normative was considered [63]: we attempted to subdivide the great number of collected data to obtain five uniform classes of wall typology (Very Low—VL, Low—L, Medium—M, High—H, and Very High—VH values of the Thermal Transmittance U).

In the same way, we proceeded for the other variables, grouping the variety of the collected examples in the classes as reported in Table 1. Once the input variables and the classes of their variability were defined, several building typologies were selected in order to guarantee a wide applicability of the automatic diagnosis procedure. In particular, buildings typical of the tertiary sector—i.e., schools, hotels, public offices, and residential and industrial buildings—were considered. Moreover, cases with as broad as possible inputs were chosen for each typology, including the marginal cases (in terms of the range of values related to each class of Table 1), for a total number of case studies of about 45. Then, all of the input parameters were estimated for these case studies (Table 2 summarizes the data related to a limited number of examples), identifying the corresponding classes among the ones in Table 1. For each building, the heating energy requirement was determined by

means of a suitable calculation program developed by the authors in accordance with UNI-TS 11300 [64]. It is possible to observe that the warehouse has low thermal transmittance for walls and high values for the glazing; despite the WWR being very low (<6%), the mean value of the winter energy demand is Very High (VH). All of the types of building chosen for the analysis present poor thermal performance and, consequently, high values of winter energy demand (High, Very High, or Very Very High). Only the public building, characterized by the largest glazing surfaces in comparison with the opaque ones (very high WWR), has a lower winter energy demand, thanks to the good performance of the walls and the roof, and the medium thermal transmittance values for the glazings (about 1.4 W/m$^2$K).

**Table 2.** Values of the input variables for the different types of buildings analyzed (VVL = very very low, VL = very low, L = low, M = medium, H = high, VH = very high, VVH = very very high).

| Case Study | Uwall (W/m$^2$K) | | Uroof (W/m$^2$K) | | Uglass (W/m$^2$K) | | WWR (%) | | WinterEnDem (kWh/m$^2$) | |
|---|---|---|---|---|---|---|---|---|---|---|
| Warehouse | 0.364 | L | 0.805 | H | 2.856 | H | 2.66 | VL | 96.12 | VH |
| Hotel | 1.295 | H | 1.291 | VH | 5.848 | VH | 6.73 | L | 97.91 | VH |
| Office | 1.394 | H | 1.406 | VH | 4.337 | VH | 7.22 | L | 164.44 | VVH |
| Apartment | 0.55 | M | 0.937 | H | 2.954 | H | 7.44 | L | 73.68 | H |
| Single house | 0.272 | VL | 1.15 | VH | 2.954 | H | 7.71 | L | 116.60 | VH |
| Historical house | 1.835 | VH | 0.88 | H | 5.941 | VH | 8.07 | L | 162.62 | VVH |
| School | 1.185 | H | 0.489 | L | 2.723 | H | 8.23 | L | 97.13 | VH |
| Public building | 0.21 | L | 0.315 | L | 1.399 | M | 33.44 | VH | 39.04 | L |

Table 2 shows that, for all the cases, the percentage of the transparent elements is not very high (2.66–8.33%), except for the public building (about 33%). They have poorly performing glazing systems, with U-values of about 4–6 W/m$^2$K, typical of a single layer glazed window; values of about 2.7–2.9 correspond to double glazing systems without thermal surface treatment. Finally, the apartment has a not-very-high energy demand (about 77 W/m$^2$K) thanks to a medium value of the thermal transmittance of the walls (only 0.55 W/m$^2$K). Furthermore, regarding output variables, the first step concerned their identification and the definition of the domains and fuzzy sets reproducing the variables' subdivision into classes.

As possible interventions for building performance enhancement, the insulation of opaque vertical surfaces (through a standard internal insulation system), roof insulation, and the replacement of glass surfaces with specific low-emissivity double glazing (U-value 1.43 W/m$^2$K, solar factor 0.67) were considered in the present study as the more frequent options. In particular, a coating system applied in the inside walls with a thermal effect was considered: it is composed of 3 cm of polystyrene and about 0.5 cm of wood-fiber-based gypsum plaster, for a total thermal transmittance of 0.68 W/m$^2$K. For the roof, an insulation system positioned between the existing structure and the top part was inserted. It is composed of 5 cm wood fiber-based panel (thermal conductivity 0.038 W/mK), together with a waterproof layer and a steam barrier; the new insulating package was installed under the underlay at the eaves (tiles or other types of covering materials of the roof). Finally, the glazing systems were replaced by a double glazing system, with a low-e treatment in the third face (6 mm float, 18 mm air, 8 mm low-e glazing, total thermal transmittance of 1.43 W/m$^2$K).

The opportunity levels for these intervention typologies, therefore, were chosen as the output variables of the fuzzy tool. Specifically, the opportunity-level variables for the insulation of opaque vertical surfaces, roof insulation and the replacement of glass surfaces with specific low-emissivity double glazing were named "InsulVertSurf", "InsulRoof" and "LowEmissGlass", respectively. Moreover, for each intervention (i.e., for each of the output variables), five classes, from 1 to 5, were defined for the level of opportunity (1—much necessary; 2—necessary; 3—programmable; 4—poorly necessary; 5—not necessary).

After the definition of the input and output variables, and their related classes, the associated membership functions (MFs) were set. Figure 3 shows the MFs considered for the $U_{wall}$, $U_{roof}$ and $U_{glass}$ parameters. Each function ranges between 0 and 1, depending on the transmittance values. Values greater than 0 are for MFs from "very low" to "very high" for thermal transmittance, steadily increasing from 0 to about 5–6 W/m²K (for the specific divisions in classes see Table 1, in the first, second, and third columns). The MFs defined for the WWR and the winter energy demand (WinterEnDem) are depicted in Figure 4. MFs with values greater than 0 switch from "Very Low" (values close to 1 for WWR lower than 6%) to "Very High" (values close to 1 are achieved for WWR higher than 20%), depending on the WWR value. Similarly, this occurs for the Winter Energy Demand; values close to 1 are achieved for the "Very Very Low" MF for an input parameter lower than 10 kWh/m² year, while they are achieved for "Very Very High" MF when the Winter Energy Demand exceeds 120 kWh/m² year. Finally, Figure 5 reports, as an example, the triangular MFs chosen for the output variable "InsulVertSurf", which represents the level of opportunity of opaque wall insulation varying from degree 1 to degree 5 (it is highlighted as the discretization and the relative MFs are the same for the three output variables).

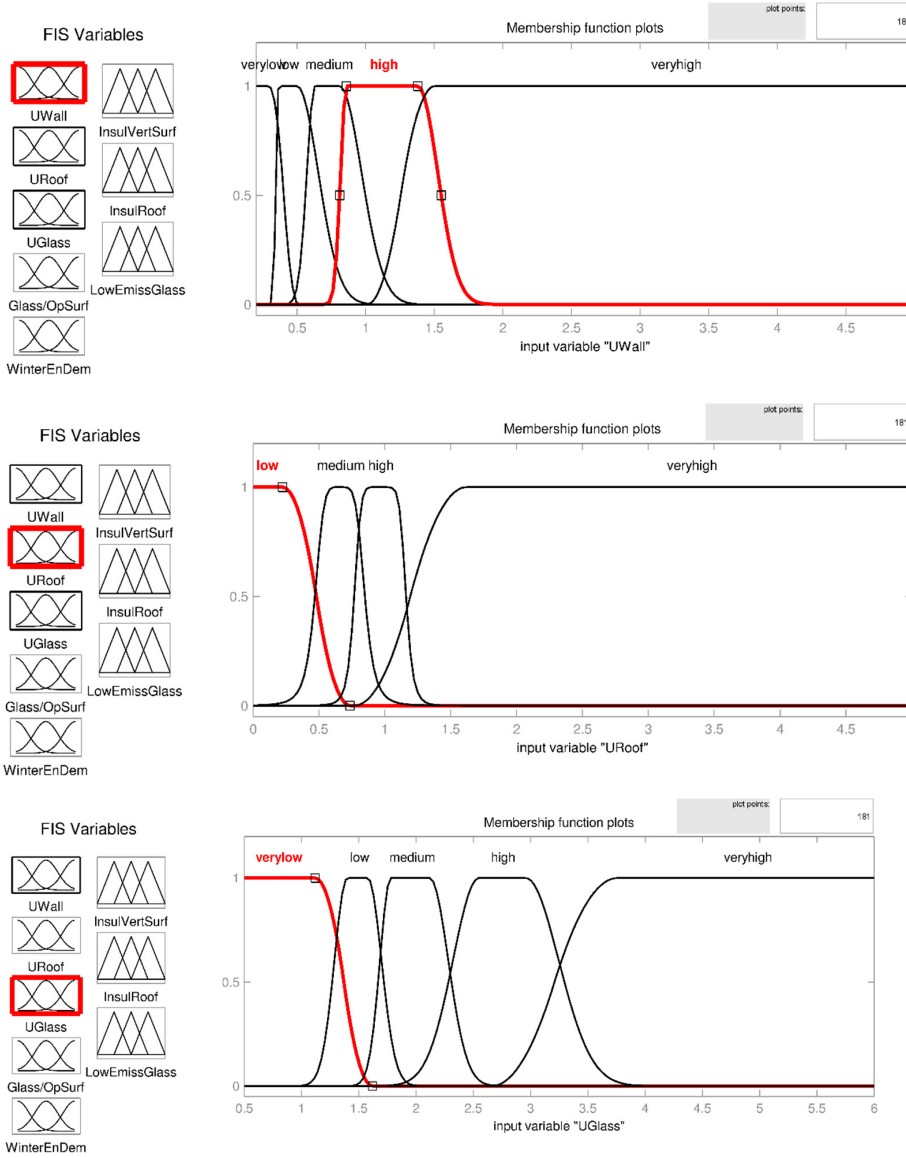

**Figure 3.** Membership functions defined in the domains of the $U_{wall}$, $U_{roof}$ and $U_{glass}$ input variables.

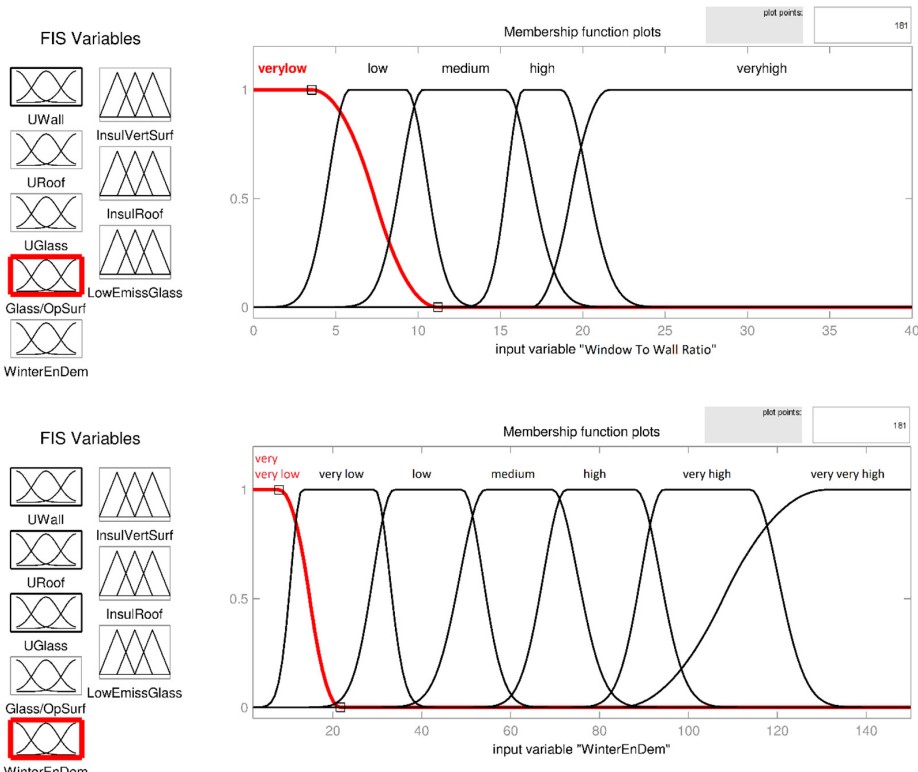

**Figure 4.** Membership functions defined in the domains of the WWR and WinterEnDem input variables.

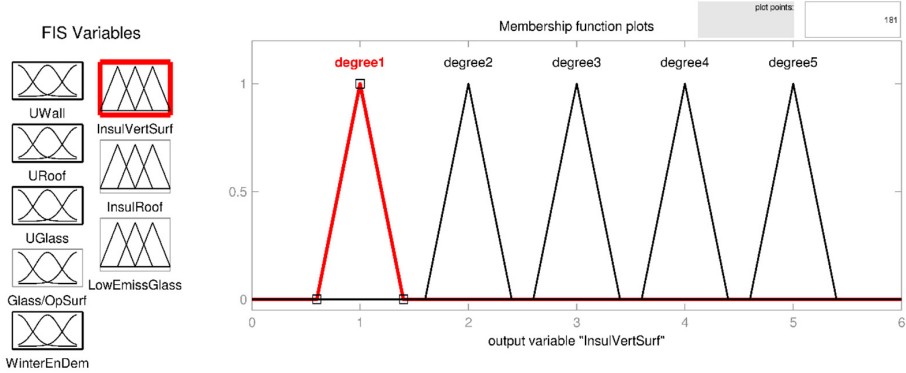

**Figure 5.** Membership functions for the output variables (insulation of vertical surfaces).

### 3.2. Design of the Inference Engine: Definition of the Rules Set

After the discretization of each variable, the different mechanisms of interaction have to be specified, defining the inference rules set. Because the task of an expert system is to make an independent decision with regard to some situations of uncertainty, not based on a deterministic model, proper rules and the reasonableness of the decisions taken by the system were defined in retrospect.

To this aim, a certain amount of data was produced by means of simulations and the evaluations of experts. In the following, some details are provided in reference to this preliminary phase. Specifically, in order to assess the actual efficiency improvements resulting from the implementation of specific interventions, and therefore their opportunity (degree from 1 to 5) for each investigated building, the winter energy savings were evaluated by means of a suitable software tool (in accordance with UNI-TS 11300 [64]). It was used to predict the winter energy savings in the case of the realization of a single intervention among the ones detailed in Section 3.1 (that is the insulation of the vertical

surface, the roof insulation and the replacing of the glazing surface with low-e solutions, and so on). Mainly on the basis of such outcomes (in terms of the required Energy Demand and the corresponding Energy Saving (in %) obtained after the application of the intervention), beyond that, the intervention typology (taking into account the related cost and technical feasibility, both quantitative and qualitative parameters), the authors assigned for all of the analyzed buildings the proper level of opportunity from 1 (low effectiveness) to 5 (high effectiveness).

Thus, a dataset constituted by experts' evaluations, made on both quantitative and qualitative criteria, was determined for a large number of buildings, each subjected to all of the investigated interventions applied singularly. Table 3 reports only some examples, for the sake of brevity, in order to highlight the several building typologies considered. It is evident, as for the warehouse, that the most convenient intervention is the insulation of the roof, whereas the requalification of the transparent surfaces is not suitable, considering the low WWR-value. For the hotel and the office types, the insulation of vertical surfaces is the best intervention (for an energy saving in the 17–20% range). The same refurbishment intervention is suggested for historical houses and schools (with the thermal transmittance of the walls being High and Very High). For the public building, the window replacement is the most suitable solution, because the walls and the roof have good thermal performance, and consequently the enhancement of opaque envelope insulation does not improve it further.

**Table 3.** Energy Saving (%) and opportunity (Opp.) of the intervention for some of the buildings.

| Case Study | Insulation of Vertical Surface | | | Low Emission Glass Surface | | | Insulation of Roof Surface | | |
|---|---|---|---|---|---|---|---|---|---|
| | Energy Demand (kWh/m$^2$) | Energy Saving (%) | Opp. | Energy Demand (kWh/m$^2$) | Energy Saving (%) | Opp. | Energy Demand (kWh/m$^2$) | Energy Saving (%) | Opp. |
| Warehouse | 92.24 | 4.04% | 4 | 94.64 | 1.54% | 5 | 75.91 | 21.03% | 1 |
| Hotel | 76.55 | 20.81% | 1 | 88.48 | 8.47% | 2 | 85.51 | 11.54% | 3 |
| Office | 136.15 | 17.20% | 1 | 154.86 | 5.83% | 2 | 126.7 | 22.95% | 1 |
| Apartment | 65.39 | 11.25% | 2 | 70.16 | 4.78% | 4 | 59.01 | 19.91% | 2 |
| Single house | 114.06 | 2.18% | 4 | 110.98 | 4.82% | 3 | 78.4 | 32.76% | 1 |
| Historical house | 98.51 | 39.42% | 1 | 151.97 | 6.55% | 2 | 157.4 | 3.21% | 4 |
| School | 72.97 | 24.87% | 1 | 96.15 | 1.01% | 5 | 95.1 | 2.09% | 4 |
| Public building | 39.04 | 0.0% | 5 | 38.99 | 4.65% | 4 | 39.04 | 0.0% | 5 |

After the preliminary phase devoted to the realization of a dataset constituted of experts' evaluations, the definition of the inference rules set was performed according to a two-steps procedure. At first (step 1), a preliminary design was carried out in order to achieve the convergence of the diagnostic tool output on the experts' evaluations. The first subjective assessment allowed us to identify and to correct the most egregious faults, so that the subsequent design step (step 2) could concentrate on performance optimization, improving a system that was already working in a substantially correct way. Therefore, additional simulations and experts' evaluations corresponding to conditions different from those considered in step 1 were defined and carried out. The information obtained during these tests was used to modify and to adapt the system, in order to make it more suitable to the task it must perform. Table 4 reports the resulting rules set, representing the knowledge base of the developed fuzzy inference system. The antecedent (the 'if' section) and the consequent (the 'then' section) of the rules describe the mode of thinking of the Fuzzy Logic system, specifically in terms of the correlation between the few considered parameters, characteristic of the envelope, and its energy performance and the degree of the effectiveness achievable thanks to each intervention. This is highlighted as, in particular, all of the rules were formulated with the unit weight and considering an 'and' logical connection for the antecedent part.

**Table 4.** Rules of the fuzzy system.

If ($U_{Wall}$ is high) and ($U_{Roof}$ is high) and ($U_{Glass}$ is high) and (WWR is medium) and (WinterEnDem is very very high) then (InsulVertSurf is degree1)(InsulRoof is degree3)(LowEmissGlass is degree2)

If ($U_{Wall}$ is high) and ($U_{Roof}$ is high) and ($U_{Glass}$ is veryhigh) and WWR is low) and (WinterEnDem is very high) then (InsulVertSurf is degree1)(InsulRoof is degree3)(LowEmissGlass is degree2)

If ($U_{Wall}$ is veryhigh) and ($U_{Roof}$ is high) and ($U_{Glass}$ is veryhigh) and (WWR is low) and (WinterEnDem is very very high) then (InsulVertSurf is degree1)(InsulRoof is degree4)(LowEmissGlass is degree2)

If ($U_{Wall}$ is veryhigh) and ($U_{Roof}$ is high) and ($U_{Glass}$ is veryhigh) and (WWR is high) and (WinterEnDem is very very high) then (InsulVertSurf is degree1)(InsulRoof is degree2)(LowEmissGlass is degree1)

If ($U_{Wall}$ is low) and ($U_{Roof}$ is high) and ($U_{Glass}$ is high) and (WWR is verylow) and (WinterEnDem is very high) then (InsulVertSurf is degree4)(InsulRoof is degree1)(LowEmissGlass is degree5)

If ($U_{Wall}$ is verylow) and ($U_{Roof}$ is high) and ($U_{Glass}$ is high) and (WWR is low) and (WinterEnDem is very high) then (InsulVertSurf is degree4)(InsulRoof is degree1)(LowEmissGlass is degree3)

If ($U_{Wall}$ is low) and ($U_{Roof}$ is high) and ($U_{Glass}$ is high) and (WWR is medium) and (WinterEnDem is high) then (InsulVertSurf is degree2)(InsulRoof is degree3)(LowEmissGlass is degree3)

If ($U_{Wall}$ is medium) and ($U_{Roof}$ is high) and ($U_{Glass}$ is high) and (WWR is low) and (WinterEnDem is high) then (InsulVertSurf is degree2)(InsulRoof is degree2)(LowEmissGlass is degree4)

If ($U_{Wall}$ is high) and ($U_{Roof}$ is low) and ($U_{Glass}$ is high) and (WWR is low) and (WinterEnDem is very high) then (InsulVertSurf is degree1)(InsulRoof is degree4)(LowEmissGlass is degree5)

If ($U_{Wall}$ is low) and ($U_{Roof}$ is low) and ($U_{Glass}$ is medium) and (WWR is veryhigh) and (WinterEnDem is low) then (InsulVertSurf is degree5)(InsulRoof is degree5)(LowEmissGlass is degree4)

If ($U_{Wall}$ is verylow) and ($U_{Roof}$ is low) and ($U_{Glass}$ is verylow) and (WWR is veryhigh) and (WinterEnDem is low) then (InsulVertSurf is degree5)(InsulRoof is degree5)(LowEmissGlass is degree5)

If ($U_{Wall}$ is high) and ($U_{Roof}$ is high) and ($U_{Glass}$ is veryhigh) and (WWR is low) and (WinterEnDem is very very high) then (InsulVertSurf is degree1)(InsulRoof is degree1)(LowEmissGlass is degree2)

For the aggregation step, which combines the outputs of all of the activated rules to give rise to a single output fuzzy set, the maximum method was applied. Figure 6 shows an example of an operating point where the rules currently enabled are highlighted; the related system response is also visible (levels of opportunity of 2, 2 and 4, respectively, for the insulation of opaque vertical surface, insulation of the roof, and low-emission glasses).

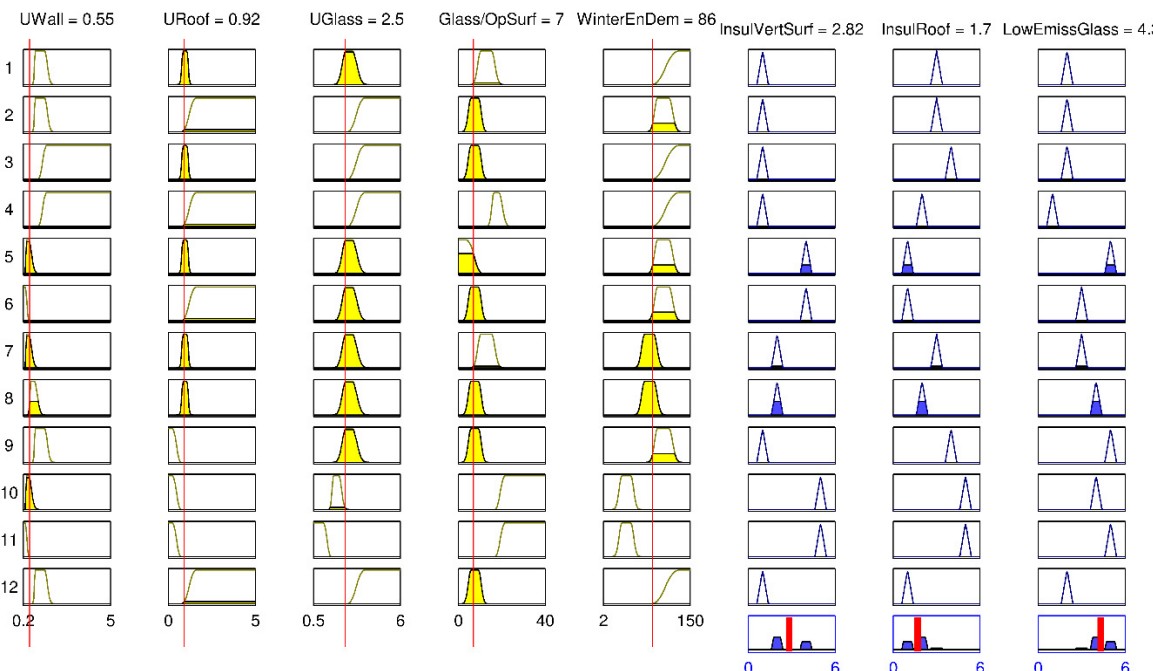

**Figure 6.** Enabled rules for a particular case study.

## 4. Validation of the Developed Tool

Finally, the tool was validated considering buildings not used in the development step, for a total of 20 cases. In particular, Table 5 summarizes—for brevity—only five examples (each related to a different building typology), which were chosen because they are representative of the tool performance. By analyzing these data, the agreement is evident between the degrees of opportunity initially attributed by the authors (i.e., the experts' evaluation) through suitable evaluations (energy saving, intervention cost and feasibility) and the ones supplied by the developed tool. In particular, the best correspondence was obtained for warehouses and schools, as shown in Table 5. These results validate its good performance, and beyond that the effectiveness of the methodology proposed concerning the application of fuzzy logic techniques.

**Table 5.** Results for the validation of the developed tool (VL = very low, L = low, M = medium, H = high, VH = very high).

| Case Study | $U_{wall}$ | $U_{roof}$ | $U_{glass}$ | WWR | Energy Demand | Opp. Degree for Opaque Surface Insulation | | Opp. Degree for Roof Insulation | | Opp. Degree for Low Emission Glass | |
|---|---|---|---|---|---|---|---|---|---|---|---|
| | | | | | | By Authors | By Tool | By Authors | By Tool | By Authors | By Tool |
| Apartment | H | H | H | L | H | 2 | 1.82 | 2 | 2.16 | 4 | 3.62 |
| Hotel | H | H | H | L | H | 1 | 1.32 | 3 | 2.62 | 3 | 2.61 |
| Warehouse | VL | H | H | VL | VH | 4 | 4.02 | 1 | 1.02 | 3 | 3.00 |
| School | L | H | H | H | VH | 2 | 1.98 | 3 | 3.00 | 3 | 3.01 |
| Historical house | H | H | H | L | VH | 1 | 1.18 | 3 | 2.73 | 2 | 2.18 |

## 5. Conclusions

This paper addresses the development of an automatic evaluation system, based on a fuzzy logic technique, to support the best choice of energy saving measures to be applied to existing buildings. The tool was developed considering winter heating, considering some elements of the building envelope, and it was successfully validated in order to support the effectiveness of the presented methodology. The main outcome of this paper is a new methodology proposed for the development of decisional tools for the energy refurbishment of buildings. It represents a relevant innovation with respect to the state of the art, with a potential significant impact on the operators (e.g., designers) of the building retrofitting field. The proposed system, which is suitable for the analysis of the building sector, reduces the uncertainty of the decision about the opportunity of some of the main interventions and produces a strong reduction in the expert evaluation effort.

The initial part of the paper reports a short description of the main elements of fuzzy logic. Then, the design of the automatic system for the evaluation of the best energy saving measures is described. The design of the fuzzy decisional tool starts from the analysis of the main elements responsible for the greater heat losses during the winter season, and the consequent identification of the priority of the useful interventions. This work represents only the initial step in the research for a tool which is able to solve complex and different problems with a single informatics tool and a single language. The developed tool was successfully tested on buildings which were different from the ones considered in the development phase, in order to verify the validity and the reliability of the proposed system. In the future, the developed methodology will be improved, considering for example a larger range of applications, the extension of the considered intervention typologies, and the application to the summer case. In particular, it is necessary to increase the number of cases to be considered as inputs for the network in order to train the fuzzy tool. The architectural design and the category of building have great influence on the effectiveness of the fuzzy instruments, the development of different tools for each typology will be interesting. Furthermore, the number of buildings to be tested and not used for the development of the tool should be also increased in order to verify the effectiveness of the developed tool.

The proposed methodology, rather than the tool currently developed—which, as indicated above, could be improved—can provide a significant impact in the framework of a project qualification. The reduction in the uncertainty of the decision about the opportunity among several interventions over a very wide application framework, as well as the reduction in the evaluation time, could significantly contribute to the control of the project quality in the energy building requalification. This advantage is typical in project activities when a multitude of different parameters have to be considered over a very wide set of application typologies. A previous experiment, but in a different ambit of civil engineering, was performed by one of the authors who contributed to the development of an expert system applied to the evaluation of the rehabilitation quality following the 1997 Umbria earthquake [65].

**Author Contributions:** Conceptualization, L.B.; methodology, E.B. and L.B.; formal analysis, G.B. and C.B.; data curation, E.M.P.; writing—original draft preparation, L.B. and E.M.P.; writing—review and editing, C.B. and E.B.; visualization, G.B.; supervision, L.B. All authors have read and agreed to the published version of the manuscript.

**Funding:** This research received no external funding.

**Institutional Review Board Statement:** Not applicable.

**Informed Consent Statement:** Not applicable.

**Data Availability Statement:** Further data presented in this study are available on request from the corresponding author. For the sake of brevity, some data are not publicly available.

**Conflicts of Interest:** The authors declare no conflict of interest.

### Nomenclature

| | |
|---|---|
| InsulVertSurf | output variable relative to the insulation of the vertical opaque surface |
| Window to Wall Ratio WWR | percentage of the glass area to the opaque |
| InsulRoof | output variable relative to the insulation of the horizontal opaque surface |
| LowEmissGlass | output variable relative to a low-emission glass surface |
| u | fuzzy set variable |
| $U_{wall}$ | thermal transmittance of the vertical opaque surface, $W/(m^2\ K)$ |
| $U_{glass}$ | thermal transmittance of the glass surface, $W/(m^2\ K)$ |
| $U_{roof}$ | thermal transmittance of the horizontal opaque surface, $W/(m^2\ K)$ |
| WinterEnDem | winter energy demand, $kWh/m^2$ year |
| μ | membership function |

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
