# Peer review of "Development of a Decisional Procedure Based on Fuzzy Logic for the Energy Retrofitting of Buildings"

_sustainability, doi:10.3390/su13169318_

Round 1

Reviewer 1 Report

The authors have conducted an interesting research on developing an automatic fuzzy-based tool to identify proper solutions for energy retrofitting of existing buildings. It requires some improvements. My comments are as follows.

In the abstract, highlight the gap in research clearer and the necessity for developing such a tool.

In the introduction, the first paragraph should be revised; probably should be merged with the subsequent paragraph.

Overall, avoid writing a very short paragraph. Each paragraph must have an introduction, body, and conclusion. Please revise the whole manuscript.

The introduction is long. Authors may consider splitting it into two sections or add sub-sections where the literature is reviewed.

Lines 188-192: remove the bullet points and mention them in the text.

Figs.1 and 2, the quality can be improved. Are they redesigned by the authors? If yes, you can mention it in the caption.

Section 3.1. remove the bullet points and mention them in the text

In the conclusion, you can write it in two or three paragraphs. Also, mention the academic and practical implications clearly.

Reviewer 2 Report

It is understood that this paper presents research results on the development of a fuzzy logic-based automation system that can quickly make design decisions in retrofit buildings considering the cost as well as heating energy savings. Despite a very long and complex literature review and detailed descriptions of the fuzzy theory and process, it is judged that the composition, development method, and explanation of the results of the fuzzy tool are not clear. Therefore, it was judged that the overall explanation method of the paper needed to be sufficiently improved. The detailed descriptions for the judgment is as follows.  

1. Literature Review This paper describes 71 references until the line number 127. It is recommended to read and organize as many relevant literature as possible. However, this paper includes references that deal with too broad a subject, so the message to be conveyed through the literature review is not clear. For example, the topics covered by the references are too diverse, such as load calculation, consumption estimation, and comfort estimation. The description of the evaluation method is also diverse, such as statistical methods, engineering analysis, and ANN. However, in the end, the message the author wants to convey through the literature review is that there are not many cases of developing a relatively simple design support tool necessary for decision making in a retrofitting project. In addition, the analysis of research cases on fuzzy logic, the method used in this paper, is very short. Therefore, it is necessary to briefly explain the contents that are not directly related to these research contents, and to add a review of the existing research literature on fuzzy logic, the research topic of this paper. Otherwise, it is difficult to judge how differentiated and progressive the research content presented in this paper is compared to existing similar studies. Also, most of the literature reviews refer to documents that are too old. If possible, a literature review based on recent research trends and the latest research contents should be added.  

2. Research objectives The purpose of the study is to select energy-saving measures, but it is also explained that the decision is made in consideration of cost and feasibility qualitative criteria. However, it is difficult to find an explanation of the process in which this part is included in the results anywhere in the paper. I hope that the authors add a description or supplement this part to make it easier to find and understand.  

3. Fuzzy Decision Tool Chapter 3 seems to be the most important part. However, it is difficult to understand specifically how the tool is made and configured. Through Tables 1 and 2, it is possible to understand roughly how input variables are composed and applied by building usage. However, the explanation of how to understand Figures 3, 4, and 5 is too short, and the results shown in Tables 3 and 4 are too short to know by what process it was made. I think that a more specific and easy-to-understand explanation of the tool is needed first.

Round 2

Reviewer 1 Report

The authors have addressed my comments and the manuscript is ready for publication after a minor correction.

as mentioned earlier, please avoid writing a very short paragraph. Each paragraph must have an introduction, body, and conclusion. Please revise the whole manuscript. (e.g. lines 216-217, 257-259, 275-278, etc.)

Author Response

Thanks for your comment. All the short paragraphs were erased and the structure you suggested was followed.

Reviewer 2 Report

Most of descriptions by the reviewer are addressed clearly and deeply.

Author Response

Thanks for your comment. Best regards.